# Effect of fluid intake on cognitive function in older individuals: A prospective study

**Hideyuki Hoshi[1], Yusuke Kakubari[2], Emi Moriya[3], Keita Shinada[2], Yoshihito Shigihara[1]\***

**1** Precision Medicine Centre, Hokuto Hospital, Obihiro City, Hokkaido, Japan, **2** Geriatric Health Services Facility Kakehashi, Hokuto Hospital Group, Obihiro City, Japan, **3** Clinical Laboratory, Hokuto Hospital, Obihiro City, Hokkaido, Japan

\* y-shigihara@hokuto7.or.jp

## Abstract

### Background

Adequate fluid intake is essential for maintaining cognitive health in older adults. However, two key questions remain unanswered before recommending increased fluid consumption: (1) whether the relationship between fluid intake and cognitive improvement is linear and (2) the underlying mechanisms that mediate this association.

### Methods and Findings

Thirty-three older adults residing in a geriatric health service facility and receiving nursing care were enrolled in this study. Fluid intake was recorded as part of routine clinical practice. Cognitive function was assessed twice during their stay using the Japanese version of the Mini-Mental State Examination (MMSE-J). Additionally, cerebral blood flow was evaluated bilaterally in the common carotid arteries using ultrasonography, with assessments conducted approximately $82.6 \pm 14.9$ days apart. Relationships among fluid intake per lean body mass (LBM), changes in MMSE-J scores, and ultrasonographic parameters were analysed using Spearman's linear correlation analysis with non-parametric bootstrapping. Correlation analyses revealed a positive linear association between fluid intake and improvement in MMSE-J scores [$P$(FDR) = 0.012] when the intake was less than 42 mL/LBM (kg) per day. Furthermore, fluid intake was inversely correlated with the resistance index in the right common carotid artery [$P$(FDR) = 0.046], indicating altered cerebral blood dynamics. The main limitations of our study include (1) the inability to evaluate baseline hydration status or fluid intake prior to facility admission due to clinical constraints and (2) the observational design precluding causal inference between fluid intake, cognitive changes, and cerebral blood flow parameters.

**Data availability statement:** All data generated or analysed in this study are openly available in Mendeley Data at Shigihara, Yoshihito (2024), "Fluid intake and cognition", Mendeley Data, V1, doi: 10.17632/pbj9vcwhfg.1.

**Funding:** The author(s) received no specific funding for this work.

**Competing interests:** YS is leading a joint research project with RICOH Co., Ltd. and Itoen Co., Ltd. HH was employed by RICOH Co., Ltd. YK, EM, and KS declare no potential conflict of interest. This does not alter our adherence to PLOS ONE policies on sharing data and materials.

## Conclusions

Within moderate intake ranges, fluid consumption was linearly associated with cognitive improvement, an effect that appears to be mediated by changes in cerebral haemodynamic.

## Introduction

Cognitive impairment, such as dementia, represents a significant concern among older adults [1]. Such impairment adversely affects daily functioning and reduces quality of life [2,3]. Although typically progressive over time [4,5], cognitive decline is not always irreversible [6,7], as it can be influenced by various modifiable lifestyle factors [8–11]. Identifying optimal lifestyle practices for preserving cognitive function is therefore essential. Among these factors, fluid intake (FI)—including the consumption of water—has been recognised as a key determinant of cognitive health. Previous studies have shown that dehydration (*i.e.,* a deficit in total body fluid [12–15]) negatively impacts cognitive performance [14,16–19], whereas maintaining adequate dehydration supports cognitive function. However, all older adults are particularly susceptible to dehydration [12,20] due to age-related physiological changes [12,19,21]. These include impaired regulation of water balance, reduced insensitivity to thirst, decreased total body water reserves, and the use of medications with diuretic properties [12,15,19,22]. The prevalence of dehydration among community-dwelling older adults ranges from 1% to 60% [20]. Dehydration events occurred in 31% of nursing home residents over 6 months [23]. These findings suggest that dehydration is both common and consequential in this population, contributing to cognitive decline that may otherwise be preventable. Therefore, encouraging adequate fluid intake among older adults remains essential. The European Food Safety Authority (EFSA) recommends a total daily water intake of 2,000 mL per day for women and 2,500 mL per day for men derived from drinking water, beverages, and food across all age groups [24]. Given that approximately 20% of total water intake is derived from food, the recommended fluid intakes from beverages alone are adjusted to 1,600 mL per day for women and 2,000 mL per day for men [12]. However, clinicians often express caution in recommending substantial increases in FI among older adults, given the prevalence of heart failure risk in this population [25–27]. Several clinical guidelines advise against excessive FI to reduce the risk of fluid overload and subsequent cardiac decompensation [28–30]. Excessive expansion of blood volume may lead to a reduction in cardiac output [31–33], and diminished cerebral perfusion resulting from impaired circulation has been associated with cognitive decline [34–36] (Fig 1B). Encouraging increased FI among older adults to support cognitive function requires evidence demonstrating an association between higher FI and cognitive improvement (Theme 1 in Fig 1) and clear explanations of the underlying mechanisms linking FI to cognitive improvement (Theme 2 in Fig 1).

This study hypothesised that adequate FI would alleviate dehydration, enhance cerebral blood dynamics, and contribute to cognitive improvement (Blue boxes in

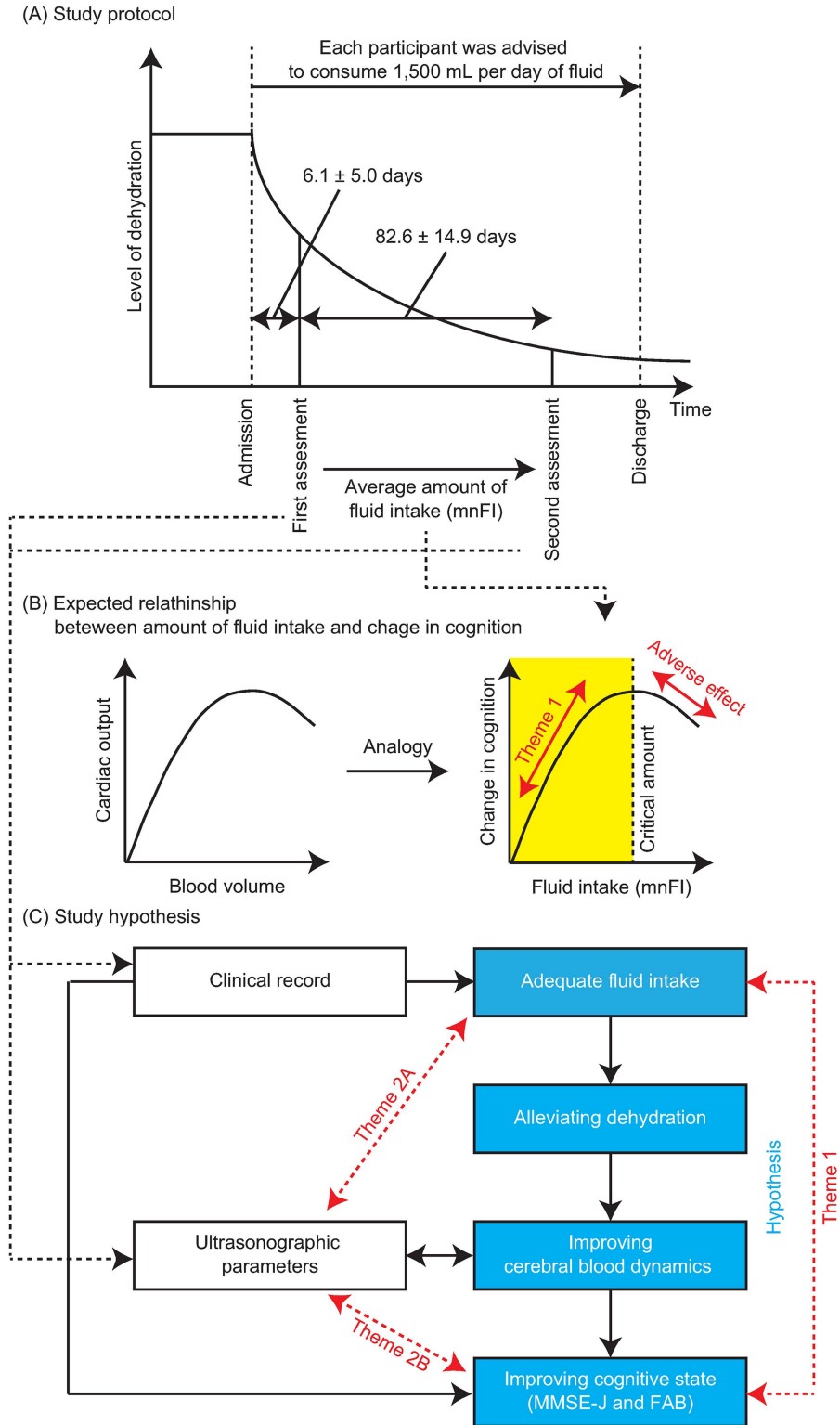

**Fig 1. Study design. (A) Study protocol.** Cognitive function and cerebral blood dynamics were assessed twice for each participant (first and second assessments). Assessment intervals varied due to clinical constraints. As part of routine clinical care, participants were advised to consume 1,500 mL of fluid per day. **(B) Expected relationship between fluid intake and cognitive change.** A positive association was anticipated between fluid intake

and cognitive improvement, provided that excessive intake does not result in adverse effects (yellow-shaded range). This expectation was based on the known relationship between blood volume and cardiac output in patients with heart failure. **(C) Study hypothesis.** Adequate fluid intake was hypothesised to alleviate dehydration, enhance cerebral blood dynamics, and promote cognitive improvement within the yellow-shaded range. Blue boxes illustrate the components of the hypothesis.

Fig 1C), provided that excessive FI does not produce adverse effects (Fig 1B). To test this hypothesis, the relationship between daily FI and changes in cognitive function was examined using clinical records from older participants residing in a geriatric health service facility (Theme 1 in Fig 1). Additionally, carotid blood flow was assessed using ultrasonography to assess cerebral blood dynamics and to investigate potential mechanisms linking increased FI with cognitive improvement (Theme 2 in Fig 1).

## Materials and methods

### Patients and ethics

Thirty-three participants (21 women; mean age±standard deviation, 86.5±8.0 years [range: 60–100 years]) who received nursing care at the geriatric health service facility 'Kakehashi' were enrolled in the present study. Participants presented with various chronic conditions: dementia (n=9), disuse syndrome (n=3), fractures (n=3), Parkinson's disease (n=2), and other individual diagnoses. Detailed diagnostic and demographic information is provided in Table 1. No additional pharmacological treatments targeting cognition were administered; in accordance with the general care approach of the Japanese geriatric health service facility, which emphasises non-pharmacological interventions. Patients who (1) were admitted to the facility to receive nursing care between 12 May 2022 and 23 May 2023 and (2) agreed to participate in the present study were analysed. Meanwhile, patients with a history of stroke were excluded. This study was approved by the Ethics Committee of Hokuto Hospital (#1099). All data used in the present study were anonymised at the initial stage of collection. Only two authors (YK and YS) retained the capacity to identify individuals through comparative tables, if required. Data were accessed for research purposes between 12 May 2022 and 2 May 2024. All protocols were performed in accordance with relevant Japanese guidelines and regulations. Written informed consent was obtained from all participants with intact cognitive function prior to enrolment. For those with cognitive impairment, consent was provided by their legal guardians (*i.e.,* their family members). Decisions made by legal guardians on behalf of the participants were fully respected. This procedure was conducted in accordance with the Ethical Guidelines for Medical, Health Research Involving Human Subjects issued by the Japanese Ministry of Education, Culture, Sports, Science, and Technology.

### Study protocol and measurements

The participants remained at the facility for several months, with the duration of stay determined by medical and social factors and relevant regulations. The present study did not influence either the length of stay or the treatments provided, including the amount of FI. Cognitive status and cerebral blood dynamics were assessed twice during the stay. The first assessment was conducted early during their stay (6.1±5.0 days after admission; range: 0–16 days), when their conditions were expected to be relatively unaffected by nursing care (Fig 1 and Table 1). The second assessment was conducted several months after admission, when the nursing care was expected to have influenced their condition. The timing of this assessment was determined based on clinical convenience. The average interval between the first and second assessments was 82.6±14.9 days (range: 34–119 days). Although the intended interval was approximately 90 days, variations occurred due to clinical limitations and constraints.

As a clinical recommendation at the facility, all older individuals staying there were advised to consume as much fluid as possible, with a target of 1,500 mL per day. Fluids (e.g., water or green tea) were provided in scaled cups by staff members (including nurses, therapists, and care workers) at the bedside and in the dining areas during meals and upon

**Table 1. Participant characteristics and assessment dates relative to admission.**

| ID | Sex | Age | Diagnosis | Ultrasonography/cognitive status | | | Fluid intake data | | |
|---|---|---|---|---|---|---|---|---|---|
| | | | | 1st | 2nd | Interval | From | To | Duration |
| 0001* | F | 94 | Dyslipidaemia | 1 | 120 | 119 | 1 | 119 | 118 |
| 0002* | F | 89 | Dementia due to Alzheimer's disease | 4 | 95 | 91 | 1 | 94 | 93 |
| 0003 | F | 84 | Dementia due to Alzheimer's disease | 7 | | | 2 | 55 | 53 |
| 0004 | M | 93 | Traumatic haemothorax | 0 | 89 | 89 | 1 | 90 | 89 |
| 0005 | F | 96 | Dementia | 4 | 100 | 96 | 1 | 104 | 103 |
| 0006 | F | 88 | Geriatric psychosis | 0 | | | 1 | 32 | 31 |
| 0007 | F | 81 | Parkinson's disease | 12 | 90 | 78 | 1 | 90 | 89 |
| 0008 | M | 83 | Cubital tunnel syndrome | 12 | 90 | 78 | 1 | 90 | 89 |
| 0009 | M | 97 | Aspiration pneumonia | 3 | 93[a] | 90 | 1 | 90 | 89 |
| 0010 | F | 84 | Dementia | 2[a] | 93 | 91 | 1 | 88 | 87 |
| 0011 | F | 90 | Hypoxemia | 1 | 86 | 85 | 1 | 88 | 87 |
| 0012 | F | 67 | Parkinson's disease | 2 | 87 | 85 | 1 | 88 | 87 |
| 0013 | M | 85 | Fracture | 16[a] | 93[a] | 77 | 2 | 87 | 85 |
| 0015 | F | 87 | Aplastic anaemia | 7 | 92[a] | 85 | 1 | 87 | 86 |
| 0016 | F | 60 | Dementia due to Alzheimer's disease | 0 | 85 | 85 | 1 | 87 | 86 |
| 0017 | M | 87 | Dementia with Lewy bodies | 6 | | | | | |
| 0018 | F | 86 | Chronic renal failure | 3 | 94 | 91 | 1 | 90 | 89 |
| 0019 | M | 85 | Brain tumour (post-operative) | 1 | 92 | 91 | 1 | 90 | 89 |
| 0020 | F | 81 | Bacterial pneumonia | 9 | 86 | 77 | 2 | 90 | 88 |
| 0021 | F | 90 | Dementia due to Alzheimer's disease | 8[a] | 89[b] | 81 | 1 | 90 | 89 |
| 0022 | M | 90 | Aspiration pneumonia, Parkinson's disease | 3 | 94[a] | 91 | 2 | 89 | 87 |
| 0023 | F | 86 | Rheumatoid arthritis | 1 | 35[a] | 34 | 1 | 43 | 42 |
| 0024 | M | 84 | Disuse syndrome | 15 | 93 | 78 | 1 | 89 | 88 |
| 0025 | M | 88 | Chronic heart failure | 12 | 105 | 93 | 1 | 89 | 88 |
| 0026 | F | 95 | Myofascial pain syndrome | 15[b] | 85 | 70 | 0 | 89 | 89 |
| 0027 | F | 100 | Disuse syndrome | 14[a] | 61[a] | 47 | 1 | 64 | 63 |
| 0028 | F | 80 | Disuse syndrome | 8 | 78 | 70 | 1 | 83 | 82 |
| 0029 | M | 94 | Fracture (post-operative) | 4 | 95 | 91 | 1 | 90 | 89 |
| 0030* | F | 92 | Fracture | 3 | 93 | 90 | 1 | 90 | 89 |
| 0031 | M | 91 | Dementia due to Alzheimer's disease | 12 | 105 | 93 | 1 | 90 | 89 |
| 0032 | M | 82 | Lower limb amputation | 10[a] | 103[a] | 93 | 1 | 90 | 89 |
| 0033 | F | 77 | Dementia, fracture | 4 | 81[a] | 77 | 1 | 81 | 80 |
| 0034 | F | 90 | Anxiety disorder | 3 | 80 | 77 | 2 | 87 | 85 |
| *M* | | 86.6 | | 6.1 | 88.9 | 82.6 | 1.13 | 85.1 | 84.0 |
| *SD* | | 7.9 | | 5.0 | 14.1 | 14.9 | 0.41 | 15.8 | 15.8 |
| MIN | | 60 | | 0 | 35 | 34 | 0 | 32 | 31 |
| MAX | | 100 | | 16 | 120 | 119 | 2 | 119 | 118 |

Unit: days; participants with an asterisk (*) in their IDs were excluded from the analysis of the changes due to being identified as outliers; [a]missing data on body fat percentage/LBM; [b]missing FAB data; empty cell indicates all relevant data are missing; mnFI, amount of fluid intake normalised using lean body mass and averaged across study period; LBM, lean body mass; FAB, Frontal Assessment Battery; *M*, mean; *SD*, standard deviation

request. The amount of fluid consumed was recorded by staff in the clinical records as part of routine clinical practice. Records were retrieved to cover the period between the first and second assessments as closely as possible (Table 1). As the impact of FI on the body varies according to individual body composition—specifically lean body mass (LBM)—FI was adjusted per participant's LBM. This adjustment reflects the fact that approximately three-quarters of the total body water is distributed within LBM [37–39]. Thus, FI was adjusted by each participant's LBM. LBM was calculated using participants' body weight and body fat percentage, each measured twice. The initial bodily profile assessment was conducted within one week of admission. Subsequent measurements were recorded monthly, with the record closest to the date of the second assessment (i.e., the day of cognitive and cerebral blood dynamics measurements) designated as the second bodily profile. Body weight and body fat percentage were obtained using a household body fat scale (HBF-306-A; OMRON Corporation, Kyoto, Japan). Daily FI was normalised using the participants' LBM, which was calculated for the first and second bodily profile assessments (LBM1 and LBM2) using the following formula: (body weight) × (100 − body fat percentage) × 0.01. Daily LBM was estimated by linearly interpolating between LBM1 and LBM2. Normalised FI (nFI) was calculated by dividing the daily FI by the LBM. The unit of nFI is mL/LBM (kg) per day, representing the amount of FI per kg of LBM per day. The nFI values were averaged across the study period to obtain the mean normalised fluid intake (mnFI), which was used for the statistical analysis. Notably, body fat percentage data were missing for five participants at the first assessment and nine participants at the second assessment. Therefore, the mnFI was not computed for 11 participants who lacked body fat percentage data in either or both assessments.

Cognitive status was assessed twice—on the first and second assessment days (Fig 1)—using the Japanese version of the Mini-Mental State Examination (MMSE-J) [40] and the Frontal Assessment Battery (FAB) [41]. The MMSE is the most commonly used tool for dementia screening [42], primarily evaluating learning and memory performance [43]. The MMSE-J is equivalent to the original English version, with official test materials obtained from an authorised vendor (Success Bell, Edajima, Japan). The FAB is a concise neuropsychological assessment specifically designed to evaluate frontal lobe function [41], where MMSE shows lower sensitivity [44]. The MMSE-J and FAB are scored on scales of 0–30 and 0–18, respectively; lower scores indicate more severe cognitive impairment in both tests. These two assessments were selected for this study for two primary reasons. First, the MMSE-J is routinely administered in the facility as a part of standard clinical practice. Although MMSE is effective for evaluating global cognition [42], it is less sensitive to certain cognitive domains, such as frontal lobe functions [44]. To address this limitation, the FAB was employed, as clinical staff at the facility are experienced in its administration. Second, the neuropsychological assessments must be concise and time-efficient, as they were conducted by therapists during their demanding clinical duties. The MMSE-J score was unavailable for one participant at the second assessment, whereas the FAB scores were missing for one and two participants at the first and second assessments, respectively, due to clinical constraints.

Cerebral blood dynamics were assessed twice— on the same day as the cognitive assessments (first and second assessment days in Fig 1)—by a clinical laboratory technician (E.M.) using one of two ultrasonography devices: the ACUSON SC2000 (Siemens Healthineers, Erlangen, Germany) or the Viamo sv7 (Canon Medical Systems Corporation, Tochigi, Japan). Ultrasonography was selected over other neuroimaging techniques, such as functional magnetic resonance imaging or magnetoencephalography, owing to its feasibility for bedside use within the facility. A previous study demonstrated that ultrasonographic parameters are associated with resting-state brain activity, which is associated with cognitive function as assessed by neuropsychological tests [18]. The data acquisition procedures adhered to the methodology described in that study [18]. Four ultrasonographic parameters were measured from the left and right common carotid arteries (CCAs): diameter of the artery (DA), peak systolic flow velocity (PSV), end-diastolic velocity (EDV), and resistance index (RI). Mean velocity and pulsatility index, although assessed in the previous study, were not measured in the present study owing to technical limitations [18]. To distinguish between the measurement sides, the parameters were denoted with the prefix l (left) or r (right), such as lPSV and rRI. Ultrasonographic measurements were not performed in the three participants during the second assessment owing to their clinical conditions.

In this manuscript, the parameters measured twice [ultrasonographic parameters (*i.e.,* cerebral blood dynamics), cognitive parameters (*i.e.,* cognitive status), and bodily profiles, except daily fluid intake] are denoted with postfixes indicating the timing of the assessments: 1 (first assessment), 2 (second assessment), and c (change between the two assessments). For example, MMSE-J1 refers to the MMSE-J score obtained at the first assessment, lPSV2 refers to the PSV in the left CCA measured at the second assessment, and MMSE-Jc refers to the changes in MMSE-J scores between the first and second assessments. Some data were missing due to clinical limitations and constrains. These include cases where the second assessment had not been completed by the end of the data collection period (23 May 2023) or cases when assessments could not be conducted due to the participants' mood or condition. However, none of the participants expressed a desire to withdraw from the study. The dignity and autonomy of participants were always respected, and missing data were accepted when necessary. All available data were included in the analyses, even when some values were missing, as each analysis was conducted independently. For example, the analyses of the relationships (1) between FI and changes in cognitive function and (2) between cognitive function and ultrasonographic parameters at the second assessment were performed separately. This approach maximised the reliability of each analysis by utilising the full extent of the available data.

## Statistical analyses

Statistical analyses were conducted using MATLAB software (MathWorks, Natick, MA, USA). The relationships among the three primary factors—FI, cognitive parameters (MMSE-J and FAB scores), and ultrasonographic parameters (DA, PSV, EDV, and RI)—were examined. The primary interest was whether these factors (e.g., FI × MMSE-Jc) are significantly corrected. To evaluate these associations, a non-parametric bootstrapping bootstrapping correlation analysis was employed, consistent with the finding of our previous study [18]. This approach offers methodological advantages over classical parametric inference methods, such as avoiding the assumption of Gaussian distributions [45]. For each variable pair, Spearman's rank correlation coefficient (*rho*) was calculated by resampling the dataset with replacement across all participants 20,000 times using the 'bootstrp' function in MATLAB. The significance level (*P*-value) was defined as the smaller percentage of bootstrap samples in which the correlation coefficient was greater or less than zero. The grand mean of the correlation coefficient (*rho*) across bootstrap iterations and the corresponding *P*-values were reported. As the analysis produced a correlation matrix, in which each statistical value was tested against the null hypothesis ($rho = 0$), the results were susceptible to an increased risk of Type I error [46]. Therefore, the *P*-values were adjusted for the false discovery rate (*FDR*) using the Benjamini–Hochberg method [47]. These correlation analyses enabled the isolation of associations among the three factors, independent of external influences such as nursing care and environmental conditions. This implies that other factors were tested as noise in the correlation analyses; however, this does not preclude their potential contribution to cognitive improvement. Furthermore, control groups are not required in correlation analyses [48,49].

The analyses were structured into three parts, corresponding to the three primary factors examined in the study: FI, cognitive parameters, and ultrasonographic parameters. First, to test the main hypothesis (Theme 1 in Fig 1)—that an adequate amount of FI leads to cognitive improvement within an optimal range—the relationships between the mnFI and changes in cognitive parameters (MMSE-Jc and FABc) were investigated. These relationships were visualised using scatter plots, which revealed three participants exhibiting markedly different patterns (*i.e.,* outliers) (Fig 2A). Consequently, their data were excluded from all subsequent statistical analyses. Following this visual inspection, the relationships between the mnFI and changes in cognitive parameters (MMSE-Jc and FABc) were examined using the non-parametric bootstrapping approach described earlier. Second, to explore the associations between FI and the four ultrasonographic parameters (Theme 2A in Fig 1), the same bootstrapping correlation analyses were applied. Third, to investigate the associations between cognitive state (*i.e.,* MMSE-J and FAB scores) and the four ultrasonographic parameters (Theme 2B in Fig 1), three subsets of bootstrapping correlation analyses were conducted: one using data from the first assessment

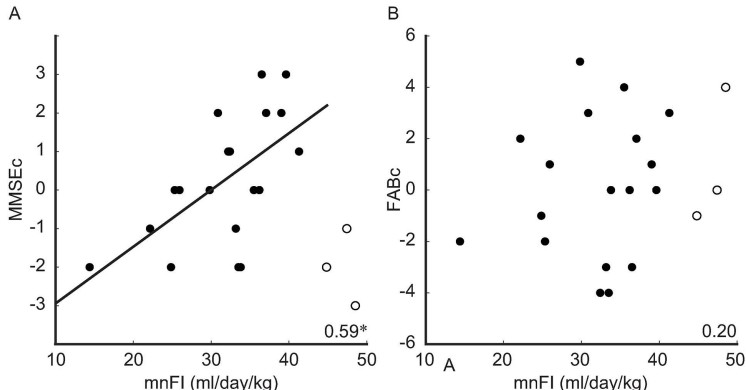

**Fig 2. Correlations between the amount of fluid intake and changes in cognitive parameters.** The value shown in the corner of each plot represents the Spearman's correlation coefficient (*rho*), averaged across bootstrap iterations; an asterisk (*) denotes a significant correlation. The filled dots indicate individual data points included in the statistical analysis, whereas circled dots denote outliers that were excluded. A least square regression line was included for significant correlations. Abbreviations: mnFI, amount of fluid intake normalised using lean body mass and averaged across the study period; MMSE-J, Japanese version of the Mini-Mental State Examination; FAB, Frontal Assessment Battery.

(e.g., MMSE-J1 × IDA1), another from the second assessments (e.g., MMSE-J2 × IDA12), and the third using the changes between assessments (e.g., MMSE-Jc × IDAc).

To provide an overview of the study context, comparisons of bodily profiles (body weight and LBM) and cognitive states (the MMSE-J and FAB scores) between the first and second assessments were conducted using non-parametric bootstrapping analyses.

## Results

### Overview of fluid intake, cognitive changes, and other profiles

The participant's average body weight measurements were 49.5 ± 10.1 kg (N = 33; range: 33.0–68.6 kg) at the first bodily profile assessment and 49.3 ± 9.5 kg (N = 33; range: 34.4–66.6 kg) at the second assessment. No significant difference was observed at the group level (P = 0.384). The average LBM measurements were 35.0 ± 6.7 kg (N = 28, range: 26.0–48.3 kg) at the first bodily profile assessment and slightly decreased to 34.7 ± 7.4 kg (N = 25, range: 26.3–54.8 kg) at the second assessment, representing a significant reduction at the group level (P = 0.045). The mnFI was 33.8 ± 8.2 mL/LBM (kg) per day (N = 22; range: 14.4–48.5 mL/LBM (kg) per day). Given the average LBM of approximately 35 kg, this corresponds to an estimated daily FI of 1,200 mL, which is lower than values recommended in the facility (1,500 mL per day) and the EFSA guideline (1,600 mL per day for women and 2,000 mL per day for men) [12,24]. In terms of cognition, the MMSE-J scores were 18.4 ± 5.0 (N = 33; range 8–29) at the first assessment and 18.2 ± 5.7 (N = 32; range 8–28) at the second assessment. The FAB scores were 8.5 ± 3.6 (N = 32; range 0–14) at the first assessment and 8.7 ± 3.5 (N = 31; range 3–15) at the second assessment. Although no significant differences were observed between the first and second assessments at the group-level (P = 0.336 for MMSE-J; P = 0.362 for FAB), individual cognitive changes varied, with 12 out of 32 participants showing improvement in MMSE-J scores and 12 out of 30 participants showing improvement in FAB scores, while others experienced declines or no change. The following section examines whether the changes in cognition scores were associated with mnFI (Theme 1).

### Associations between fluid intake and cognitive status (Theme 1)

Next, the relationships between the amount of FI (mnFI) and changes in cognitive status (Theme 1) were examined. Fig 2 presents scatter plots illustrating these relationships, which were visually inspected prior to detailed statistical

analyses. For the MMSE-J (Fig 2A), a larger amount of FI was generally associated with greater improvement in MMSE-J scores, except for three participants whose mnFI exceeded 42 mL/LBM (kg) per day (ID0001:48.52, ID0002:47.43, and ID0030:44.82; Table 1). This threshold is hereafter referred to as the 'critical amount'. Considering the average LBM of approximately 35 kg, the critical amount corresponds to roughly 1,500 mL per day. These three participants exhibited markedly decreased MMSE-J change scores (MMSE-Jc), consistent with our expectations (Fig 1B). Therefore, these participants were identified as outliers and excluded from all subsequent statistical analyses. Among the remaining participants, the mnFI was positively correlated with MMSE-Jc [N = 19, *rho* = 0.567, *P*(FDR) = 0.012]. For the FAB scores (Fig 2B), visual inspection revealed no linear association, which was confirmed by statistical analysis [N = 18, *rho* = 0.142, *P*(FDR) = 0.261].

### Associations of fluid intake with ultrasonographic parameters (Theme 2A)

To explore the potential mechanisms underlying the association between increased FI and cognitive improvement, two sets of correlation analyses were performed. First, the correlations between the amount of FI (mnFI) and cerebral blood dynamics (*i.e.,* ultrasonographic parameters) (Theme 2A in Fig 1) were investigated. The results indicated that the mnFI was positively correlated with change in the end-diastolic velocity in the right CCA (rEDVc) [N = 17 *rho* = 0.540, *P*(FDR) = 0.046] and negatively correlated with change in the resistance index measured at the right CCA (rRIc) [N = 17, *rho* = −0.537, *P*(FDR) = 0.046] (Fig 3 and Table 2). These findings suggest that increased FI was associated with elevated blood flow velocity during the end-diastolic phase and a reduction in vascular resistance. No significant correlations were found between mnFI and changes in other ultrasonographic parameters (Fig 3 and Table 2). Additionally, visual inspection of Fig 3 revealed that the three outliers (circled dots in Fig 3) exhibited patterns distinct from those of the other participants (filled dots in Fig 3) with respect to rEDVc and rRIc.

### Associations of ultrasonographic parameters with cognitive state at the first and second assessments as well as their changes (Theme 2B)

Second, the associations between cerebral blood dynamics (*i.e.,* ultrasonographic parameters) and cognitive state (*i.e.,* MMSE-J and FAB scores) (Theme 2B in Fig 1) were examined separately at the first and second assessments as well as their changes. At the first assessment, no significant correlations were observed between any ultrasonographic parameters and either the MMSE-J or FAB scores (Table 3). At the second assessment, significant negative correlations were identified between the MMSE-J score and both the lRI2 [N = 27, *rho* = −0.483, *P*(FDR) = 0.032] and rRI2 [N = 27, *rho* = −0.485, *P*(FDR) = 0.020] (Fig 4 and Table 3). The FAB score was also negatively correlated with rRI2 [N = 26, *rho* = −0.440, *P*(FDR) = 0.040]. No other significant correlations were found. In terms of the changes in parameters, no correlations were observed between cognitive scores and any ultrasonographic parameters (Table 3). Visual inspection did not indicate any differential behaviour among the three previously identified outliers. In summary, these results suggest that better cognitive performance was associated with lower vascular resistance in the microcirculation, particularly when dehydration was not severe.

## Discussion

Two principal findings emerged from the present study. First (Theme 1), a large amount of FI (mnFI) was linearly correlated with cognitive improvement, as indicated by changes in the MMSE-J score (MMSE-Jc) when the intake remained below the critical threshold of 42 mL per kg of lean body mass per day (Fig 2). Second (Theme 2), the daily amount of FI (mnFI) was significantly associated with changes in cerebral blood dynamics, as assessed by ultrasonographic parameters (Fig 3).

Dehydration is a common health issue among older adults [20] and impairs cognitive performance [12,16–20,22,23]. It alters cerebral microcirculation [16,50], hormonal levels (e.g., cortisol, serotonin, and dopamine) [51], cell metabolism

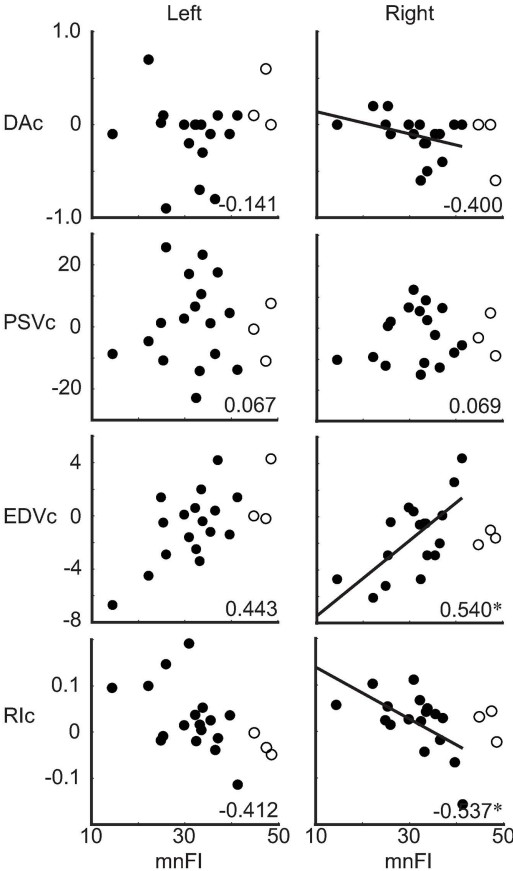

**Fig 3. Correlations between the amount of fluid intake and changes in ultrasonographic parameters.** The correlations are shown for the **(A)** left and **(B)** right common carotid arteries. The number displayed in the corner of each plot indicates Spearman's rank correlation coefficient (*rho*), averaged across bootstrap iterations; an asterisk (*) indicates significant correlation. Filled dots indicate the individual data points included in the statistical analysis, whereas circled dots indicate outliers excluded from the analysis. Least squares regression lines were added for significant correlations. Abbreviations: DA, diameter of the artery; PSV, peak systolic velocity; EDV, end-diastolic velocity; RI, resistance index; mnFI, amount of fluid intake normalised by lean body mass and averaged across the study period; CCA, common carotid artery.

**Table 2. Correlations between fluid intake and ultrasonographic parameters.**

| | | DAc | PSVc | EDVc | RIc |
|---|---|---|---|---|---|
| Left | N | 17 | 17 | 17 | 17 |
| | *rho* | −0.141 | 0.067 | 0.443 | −0.412 |
| | *P*(FDR) | 0.395 | 0.395 | 0.110 | 0.110 |
| Right | N | 17 | 17 | 17 | 17 |
| | *rho* | −0.400 | 0.069 | 0.540 | −0.537 |
| | *P*(FWE) | 0.081 | 0.379 | 0.046* | 0.046* |

DA, diameter of the artery; PSV, peak systolic flow velocity; EDV, end-diastolic velocity; RI, resistance index; N, number of datasets used in the analysis; *rho*, Spearman's rank correlation coefficient; *P*(FDR), *P*-value adjusted for the false discovery rate. An asterisk (*) indicates a significant correlation.

Table 3. Correlations between cognitive state and ultrasonographic parameters.

| | | | Left | | | | Right | | | |
|---|---|---|---|---|---|---|---|---|---|---|
| | | | DA | PSV | EDV | RI | DA | PSV | EDV | RI |
| First assessment | MMSE-J | N | 30 | 30 | 30 | 30 | 30 | 30 | 30 | 30 |
| | | *rho* | 0.059 | 0.038 | 0.316 | −0.245 | −0.074 | −0.126 | 0.217 | −0.365 |
| | | P(FDR) | 0.435 | 0.435 | 0.218 | 0.218 | 0.350 | 0.337 | 0.186 | 0.076 |
| | FAB | N | 29 | 29 | 29 | 29 | 29 | 0.50865 | 29 | 29 |
| | | rho | −0.161 | 0.149 | 0.289 | −0.216 | −0.150 | −0.029 | 0.195 | −0.219 |
| | | P(FDR) | 0.230 | 0.230 | 0.230 | 0.230 | 0.313 | 0.438 | 0.266 | 0.266 |
| Second assessment | MMSE-J | N | 27 | 27 | 27 | 27 | 27 | 27 | 27 | 27 |
| | | *rho* | −0.014 | −0.143 | 0.326 | −0.483 | −0.151 | −0.178 | 0.340 | −0.485 |
| | | P(FDR) | 0.478 | 0.333 | 0.126 | 0.032* | 0.241 | 0.219 | 0.076 | 0.020* |
| | FAB | N | 26 | 26 | 26 | 26 | 26 | 26 | 26 | 26 |
| | | *rho* | 0.090 | −0.145 | 0.178 | −0.337 | 0.020 | −0.171 | 0.261 | −0.440 |
| | | P(FWE) | 0.332 | 0.332 | 0.332 | 0.164 | 0.457 | 0.276 | 0.170 | 0.040* |
| Change | MMSE-J | N | 27 | 27 | 27 | 27 | 27 | 27 | 27 | 27 |
| | | *rho* | −0.130 | −0.226 | −0.236 | 0.034 | −0.094 | −0.322 | −0.112 | −0.053 |
| | | P(FWE) | 0.303 | 0.246 | 0.246 | 0.436 | 0.399 | 0.200 | 0.399 | 0.399 |
| | FAB | N | 25 | 25 | 25 | 25 | 25 | 25 | 25 | 25 |
| | | *rho* | −0.065 | 0.316 | 0.112 | 0.167 | 0.013 | 0.243 | 0.278 | 0.015 |
| | | P(FWE) | 0.375 | 0.292 | 0.375 | 0.364 | 0.477 | 0.288 | 0.220 | 0.477 |

DA, diameter of artery; PSV, peak systolic flow velocity; EDV, end-diastolic velocity; RI, resistance index; MMSE-J, Mini-Mental State Examination; FAB, Frontal Assessment Battery; N, number of datasets used in the analysis; rho, Sperman's linear correlation coefficient; P(FDR), P-value adjusted for the false discovery rate. An asterisk (*) indicates significant correlation.

[52,53], and synaptic structure and function [16], all of which contribute to cognitive decline [17,18]. However, the importance of maintaining adequate hydration for cognitive health has been largely overlooked, and only a limited number of studies have addressed this topic [12,19,21,53]. Notably, the World Health Organization's guidelines for preventing dementia have not yet mentioned fluid intake as a consideration [9]. To promote FI among older individuals as a means to support cognitive function, robust evidence is needed to demonstrate that adequate hydration improves cognition (Theme 1 in Fig 1) and to understand the potential underlying mechanisms (Theme 2 in Fig 1). To address these two research themes, three correlation analyses were conducted: (1) between FI and changes in cognitive state (Theme 1 in Fig 1), (2) between FI and ultrasonographic parameters (Theme 2A in Fig 1), and (3) between changes in cognitive state and ultrasonographic parameters (Theme 2B in Fig 1). Each analysis is discussed in detail below.

## Relationship between FI and change in cognitive state (Theme 1)

To address the first theme (Fig 1), the relationship between FI and changes in the cognitive state was initially examined through visual inspection of scatter plots (Fig 2) followed by statistical analyses.

A significant positive correlation was observed between mnFI and changes in MMSE-J scores (MMSE-Jc), when the mnFI remained below the critical threshold of 42 mL/LBM (kg) per day. The significant correlation was confirmed through statistical analysis [P(FDR) = 0.012]. An adequate amount of FI is likely sufficient to mitigate the participants' dehydration status, resulting in the observed cognitive changes (Fig 1C), given that a substantial proportion of older adults experience dehydration that remains undetected [20,23]. Furthermore, the absence of improvement in MMSE-J scores with excessive FI can be intuitively explained by the inverted U-shaped curve presented in Fig 2A, which resembles the relationship between cardiac output and preload (*i.e.,* blood volume) in patients with heart failure [31] (Fig 1B). With regard to the

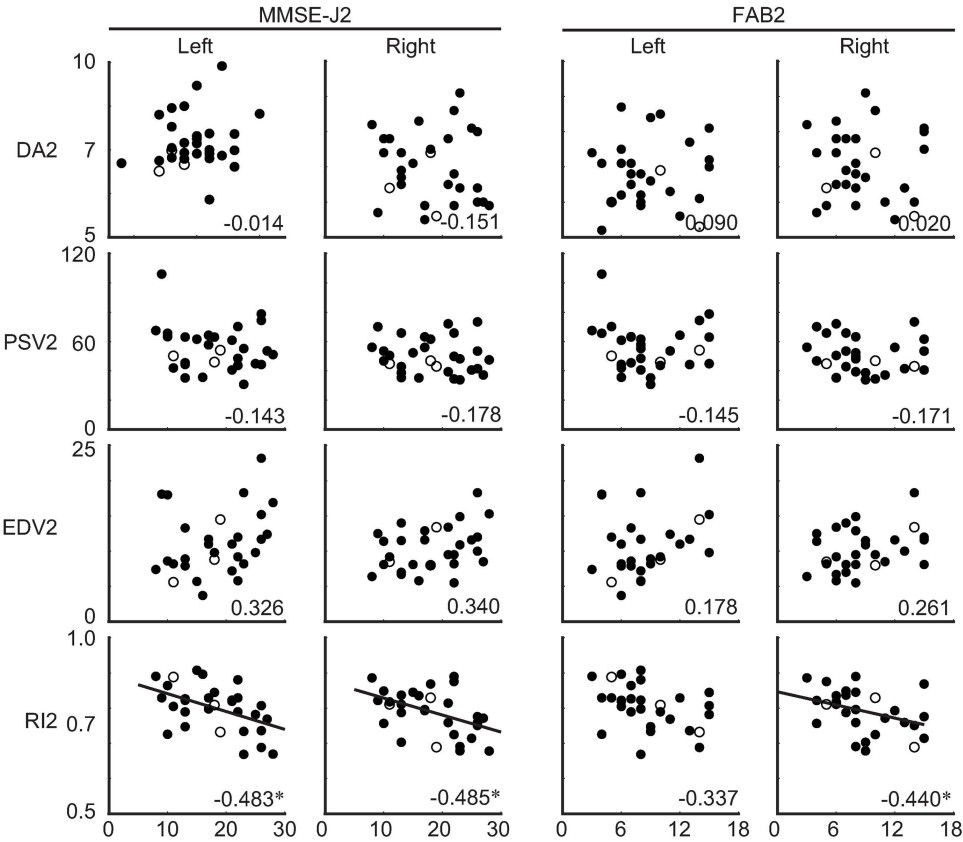

**Fig 4. Correlations between cognitive state and ultrasonographic parameters at the second assessment.** The number displayed at the corner of each plot indicates Spearman's correlation coefficient (*rho*) averaged across bootstrap iterations. Filled dots indicate the individual data considered for statistical analysis, whereas circled dots indicate the outliers excluded from the analysis. Least square regression lines were added for significant correlations. Abbreviations: MMSE-J, Japanese version of the Mini-Mental State Examination; FAB, Frontal Assessment Battery; DA, diameter of the artery; PSV, peak systolic velocity; EDV, end-diastolic velocity; RI, resistance index; CCA, common carotid artery.

FAB score, no linear association was identified between the amount of FI and FABc, as confirmed by statistical analysis [*P*(FDR) = 0.261]. Based on these findings, the amount of FI correlates positively with changes in the cognitive state with two limitations: (1) the linear association applies only within a specific range of FI, and (2) only certain cognitive domains (*i.e.,* MMSE-J, not FAB) show a correlation with FI. When encouraging older adults to increase FI to improve cognition, these potential limitations must be considered. The critical FI for the participants was identified as 42 mL/LBM (kg) per day. Given an average LBM of approximately 35 kg, the average FI was approximately 1,500 mL per day, which aligns with the facility's recommended amount (1,500 mL per day) and falls slightly below the EFSA guidelines (1,600 mL per day for women and 2,000 mL per day for men) [12,24]. The critical FI may vary among different groups of older adults and can fall below the recommended levels. Outliers with FI exceeding the critical threshold will be addressed again at the end of the Discussion section. Subsequent analyses concentrated on participants whose mnFI fell below the critical value of 42 mL per kg of LBM per day; three outliers were excluded from further consideration.

### Potential mechanisms linking FI to changes in cerebral blood dynamics (Theme 2A)

The second theme of the present study addresses the potential mechanisms underlying the relationship between increased FI and cognitive improvement (Fig 1). We hypothesised that adequate FI would alleviate dehydration, improve

cerebral blood dynamics, and lead to cognitive improvement (Blue boxes in Fig 1). To test the hypothesis, ultrasonography was employed to provide information on cerebral blood dynamics. To address Theme 2, two sets of correlation analyses were conducted: (1) between FI and changes in ultrasonographic parameters (Theme 2A in Fig 1) and (2) between cognitive state and ultrasonographic parameters (Theme 2B in Fig 1). In the first set of correlations, the mnFI was correlated with the rEDVc and rRIc (Fig 3 and Table 2). The RI represents haemodynamic resistance, primarily influenced by the distal cerebral microvascular bed [54,55]. Increased mnFI is plausibly associated with dilation of the cerebral microvessels, leading to a reduction in RI. The positive correlation observed between mnFI and EDV can be explained by the inverse relationship between RI and EDV as defined by RI = (PSV − EDV)/ PSV. A previous study has demonstrated that lower EDV predicts adverse health events, such as ischaemic attacks or ischaemic stroke [56–58]. Consequently, higher EDV reflects a healthier cerebral condition, which aligns with the finding that increased mnFI is associated with higher EDV and improved cognition. In summary, these results support the hypothesis that an adequate amount of FI (mnFI) improves cerebral blood dynamics (Theme 2A) predominantly within the cerebral microcirculation.

### Potential mechanisms linking cerebral blood dynamics to cognitive function (Theme 2B)

To confirm that cerebral blood dynamics are associated with cognitive state (Theme 2B), a second set of correlation analyses was conducted. This analysis comprised three subgroups: values obtained during the first assessment, the second assessment, and the changes between the two. The first assessment was conducted within two weeks of admission (6.1 ± 5.0 days). During this period, participants were likely affected by varying levels of dehydration, as their physical condition may have been influenced by their pre-admission lifestyles. Community-dwelling older individuals are particularly prone to dehydration (Fig 1A) [20]. Under these conditions, no significant correlation was found between ultrasonographic parameters and cognitive state (Table 3). This absence of association may be attributed to inter-individual variability in dehydration levels, which likely introduced random variances into cerebral blood dynamics and thereby weakened group-level correlations. The second assessment was conducted several months after admission (82.6 ± 14.9 days). By this time, the likelihood of dehydration had decreased, as participants had been consistently encouraged to increase their FI, with a target of 1,500 L per day (Fig 1A). Under these more stabilised conditions, MMSE-J2 scores were negatively correlated with both lRI2 and rRI2 (Fig 4 and Table 3). As previously discussed, the RI reflects resistance within the cerebral microvasculature [54,55], with higher indicating poorer microcirculatory function. These findings suggest that cognitive impairment can be partially explained by suboptimal microvascular cerebral blood flow. Cerebral blood flow is affected by hydration status [16,50], which is in turn is affected by FI. Taken together, the MMSE-J score is dependent on the condition of the cerebral microvasculature and that adequate FI contributes to cognitive improvement through enhanced microcirculatory dynamics. The FAB score was also associated with rRI2. However, no significant correlation was found between mnFI and changes in FAB scores (Fig 2B). This suggests that although the FAB score may also be influenced by cerebral microcirculation, the relationship appears to be weaker than that observed for the MMSE-J.

In the third subgroups of correlation analyses (examining the relationship between changes in ultrasonographic parameters and changes in cognitive state), no significant correlations were observed with either the MMSE-Jc or FABc scores (Table 3). These negative findings may be attributed to the absence of associations during the first assessment, likely resulting from substantial inter-individual variability in dehydration status at baseline.

### Three outliers in Fig 2

In Fig 2, three of the 22 participants exhibited divergent patterns, having consumed fluid exceeding the critical threshold [42 mL/LBM (kg) per day]. These individuals were excluded from the statistical analyses. Despite their high FI, no improvement in MMSE-J scores was observed, contrary to expectations based on fluid intake alone. Their behaviour differed from that of the others in Fig 3 (rEDVc and rRIc), whereas a similar pattern was observed in Fig 4. This suggests that the relationship between FI and cerebral blood dynamics was disrupted, whereas the relationship between cerebral blood

dynamics and cognition was preserved. This finding aligns with our expectation that excessive FI may exert an adverse effect on cardiac output, thereby negatively impacting cognitive function.

## Limitations

The current study has some limitations. First, it was an observational study, dependent on a clinical dataset recorded for care purposes. Some datasets exhibited missing values, and some minor date mismatches were observed (Table 1). However, these issues had a negligible influence on the findings, as missing data were appropriately addressed in the statistical analyses, and the study period substantially exceeded the short intervals affected by date mismatches. Additionally, causal relationships between cognitive changes and the other two factors—namely, FI and ultrasonographic parameters—could not be established, as is inherent in observational research. Second, this study was conducted at a single geriatric health service facility. Caution is therefore warranted in generalising the findings, particularly with regard to the critical FI threshold of 42 mL per kg of LBM per day, which may vary among different populations of older adults. Third, FI was not measured prior to admission, and the severity of dehydration at the time of enrollment could not be estimated. As this study was conducted in a geriatric health service facility setting, laboratory measurements such as plasma osmolality were unavailable. However, the severity of dehydration was assumed to be randomly distributed among participants, and the applied statistical methods are believed to have minimised its potential influence. The results presented here are based on statistical analyses and are reliable within the acceptable standards of scientific reporting based on *P*-values. Hence, future interventional studies should address the second and third limitations identified.

## Conclusion

Daily FI was found to influence cerebral microcirculation, leading to improvements in certain aspects of cognitive function. These results underscore the clinical significance of managing hydration status to support optimal cognitive performance, with consideration given to cerebral blood dynamics.

## Acknowledgments

We extend our deepest gratitude to the participants of this study. We would also like to express our sincere respect for their invaluable contributions to the invaluable contributions of medical science. We sincerely appreciate Dr. Hajime Kamada (Honorary chairperson, Hokuto Hospital) and Dr. Shigeru Kitamori (Chairperson, Geriatric Health Services Facility, Kakehashi) for providing access to these facilities. We appreciate the contributions and support of nurses, therapists, care workers, and all other staff in Kakehashi. We would also like to thank Editage (www.editage.com) for providing English language editing assistance.

## Author contributions

**Conceptualization:** Yoshihito Shigihara.

**Data curation:** Yusuke Kakubari, Emi Moriya, Keita Shinada.

**Formal analysis:** Hideyuki Hoshi, Yoshihito Shigihara.

**Investigation:** Hideyuki Hoshi, Yoshihito Shigihara.

**Methodology:** Hideyuki Hoshi, Yoshihito Shigihara.

**Project administration:** Yoshihito Shigihara.

**Resources:** Yoshihito Shigihara.

**Software:** Hideyuki Hoshi, Yoshihito Shigihara.

**Supervision:** Yoshihito Shigihara.

**Validation:** Hideyuki Hoshi, Yoshihito Shigihara.

**Visualization:** Hideyuki Hoshi, Yoshihito Shigihara.

**Writing – original draft:** Hideyuki Hoshi, Yoshihito Shigihara.

**Writing – review & editing:** Hideyuki Hoshi, Yoshihito Shigihara.

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
