## [Decision Letter · Decision Letter 0]

15 Apr 2025

PONE-D-25-10914Fluid intake modifies cognitive status in older individuals: a prospective studyPLOS ONE

Dear Dr. Shigihara,

Thank you for submitting your manuscript to PLOS ONE. After careful consideration, we feel that it has merit but does not fully meet PLOS ONE’s publication criteria as it currently stands. Therefore, we invite you to submit a revised version of the manuscript that addresses the points raised during the review process.

Based on the reviewers' suggestions, the paper needs major revision. The reviewers' comments can be found below.

We look forward to receiving your revised manuscript.

Kind regards,

Tanja Grubić Kezele, Ph.D., M.D.

Academic Editor

PLOS ONE

Journal Requirements:

2. Thank you for stating the following in the Competing Interests section: [YS is leading a joint research project with RICOH Co., Ltd. and Itoen Co., Ltd. HH was employed by RICOH Co., Ltd. YK, EM, and KS declare no potential conflict of interest.].

We note that you received funding from a commercial source: [RICOH Co., Ltd]

Reviewers' comments:

Reviewer's Responses to Questions

**Comments to the Author**

1. Is the manuscript technically sound, and do the data support the conclusions?

Reviewer #1: Partly

Reviewer #2: No

2. Has the statistical analysis been performed appropriately and rigorously? 

Reviewer #1: No

Reviewer #2: No

3. Have the authors made all data underlying the findings in their manuscript fully available?

Reviewer #1: Yes

Reviewer #2: Yes

4. Is the manuscript presented in an intelligible fashion and written in standard English?

Reviewer #1: Yes

Reviewer #2: Yes

5. Review Comments to the Author

Reviewer #1: I commend the authors for their hard work and dedication to the implementation of this study. Please see comments below on suggestions for improving your manuscript.

Abstract

The abstract is lengthy but is lacking in key aspects. Primarily of concern, there is no actual data given for results. This is of absolute importance and the key statistical analyses and numerically reported outcomes must be presented.

Introduction

The second and third sentence begin with the word "It". Be more direct in your writing style.

Line 62: Should be no y after thirst.

You need to develop a proper problem statement.

Line 66: Your hypothesis statement occurs before your purpose statement.

Methods

My main concern is the design of the study.

It is never explicitly stated how fluid intake was measured. For example, did patients drink out of metered containers and the change in mass or volume was assessed?

You have a within subjects design, but there seems to be no true control and intervention arm. First assessment was taken between 0-16 days after admission. Then, a second assessment occurred approximately 83 days later. The induction of blinded, reduced or increased fluid intake after a few weeks of hospitalization would have been a much stronger design. The ultrasonographic score changes with reduced or increased FI could have been confirmed, and then you could even have simply tested cognition during periods of lower and higher cererbral blood flow dynamics parameters.

No background or explanation of the cognitive status questionnaires is given.

Number of participants: There are so many missing pieces of data it is difficult to follow the paper. All participants with missing data need to be removed. The abstract makes it appear that the n is almost double of the participants applicable in one of your most important figure (i.e. Figure 1) when the 3 outliers were removed. I am also not convinced the rationale for removing this data is valid.

No validation for the instrument used for determining BFP is provided. Did you see high variance in the patients? I may have overlooked where this outcome's data are located, but if no, why not just base relative fluid intake off body mass?

There is just an overload of tables and figures. It is difficult to follow.

Reviewer #2: Shigihara et al. conducted a study involving 33 Japanese elderly people. The aim of the study was to evaluate the associations between fluids intake and cognitive performances. More specifically, they assessed cerebral blood dynamics. The authors concluded that elderly people with higher fluid intake showed more improvements in their cognition, which is based on the changes in cerebral blood dynamics. Cerebral blood dynamics can be used to monitor patients’ dehydration states, which affect their cognitive status.

Major comments.

The sample size is only 33 individuals, and they are all from a single geriatric health service facility. The sample has a high degree of homogeneity, which may lack broad representativeness. As a result, it is difficult to extrapolate the research findings to a more general elderly population. It is recommended that in future studies, the sample size be expanded and elderly people from different regions and with diverse life backgrounds be included. Furthermore, It is impossible to assess the patients' fluid intake and dehydration status before admission, which may lead to biases in the research results. Although fluid intake was recorded during the study period, the situation before admission is unknown, which may interfere with the judgment of the relationship between fluid intake and cognition.

Followed are the details.

1.     Abstract: The characteristics of the survey subjects is not well defined. It is unknown whether they are healthy, and if they have chronic diseases. During the research process, it is unclear how many subjects withdrew, and for what reasons. Also, the average age of the subjects and the duration of the study are not provided. The authors just showed the “several months”, three months? Six months? Please clarify.

2.The study lack of Control Group: This study lacks a control group, making it difficult to determine that changes in cognitive status are solely due to fluid intake. For example, during the intervention period, factors such as changes in nursing methods and living environment may also affect cognitive status, but it is impossible to distinguish their effects from those of fluid intake in this study. It is recommended to add a control group in future research, such as setting parallel groups with different fluid intake standards, or comparing two groups of elderly people receiving regular care and enhanced fluid intake care, so as to more accurately assess the impact of fluid intake on cognitive status.

3.As for the methods measuring the cognitive performances, the MMSE-J and FAB were used to assess cognitive status, which may not fully reflect all aspects of cognitive function. Aspects such as executive function and other different dimensions of memory were not fully covered in this study. As shown in previous studies, the Short-term memory and attention are the most vulnerable to the effects of dehydration. But in this study, the aspects of the cognitive performances were not assessed. Moreover, when assessing brain activity, relying solely on several parameters measured by carotid ultrasound is difficult to comprehensively reflect the complex hemodynamic changes in the brain. Techniques such as functional magnetic resonance imaging (fMRI) and transcranial Doppler ultrasound (TCD) can be considered to obtain richer brain hemodynamic data.

4.Statistical Analysis: The characteristics of the elderly people was not displayed in this part, such as the gender, age, height, weight, and if they were healthy or had some chronic diseases. I recommend the authors to add the information. When dealing with multivariate relationships, using only Pearson correlation analysis and bootstrapping methods is somewhat insufficient. In the case of multiple confounding factors such as age and underlying diseases, multiple linear regression analysis should be considered to clarify the independent contributions of various factors to changes in cognitive status. Data Normality Test: The article does not mention conducting normality tests on the data, and some statistical methods (such as Pearson correlation analysis) have certain requirements for data distribution. If the data do not meet the normality assumption, it may lead to biases in statistical results. It is recommended to supplement data normality tests and select appropriate statistical methods according to the data distribution, such as non - parametric tests.

5.Results Section: In the "Results" section, the description of some key results is not detailed enough. For example, when referring to the improvement of participants' cognitive status, it only states that "More than one - third of the participants showed an increase in MMSE-J scores after the study period", without specifying the specific proportion and number of people, which is not conducive to readers' accurate understanding of the results. Specific proportion and number information should be supplemented to enhance the persuasiveness of the results. When describing the relationship between ultrasonic parameters and cognitive status, only correlation coefficients and P - values are listed, lacking an explanation of the practical significance. For example, when stating that MMSE-J2 is negatively correlated with lPI2, lRI2, and rRI2, it should be further explained what this negative correlation means clinically and how it actually affects the cognitive function of the elderly.

6.References: The number of references in this article is too small, and there are only seven references from the past five years (7/34). It is recommended that the author conduct a new literature review.

This article needs to be revised. I do not recommend its publication.

6. PLOS authors have the option to publish the peer review history of their article (what does this mean? ). If published, this will include your full peer review and any attached files.

**Do you want your identity to be public for this peer review?** For information about this choice, including consent withdrawal, please see our Privacy Policy .

Reviewer #1: No

Reviewer #2: No

---

## [Author Response · Author response to Decision Letter 1]

2 Jun 2025

Response to reviewers:

To the editor and reviewers:

Thank you for the opportunity to submit a revised version of our manuscript entitled ‘Effect of fluid intake on cognitive function in older individuals: A prospective study’.

We appreciate the time and effort you have invested in reviewing our manuscript. Your insightful comments and constructive feedback have been invaluable in enhancing the quality and clarity of our study. We have carefully revised the manuscript to address all of your suggestions and concerns. We displayed all changes in blue font instead of using the Track Changes function in MS Word, as there were too many edits in the 'Revised Manuscript with Track Changes.docx' file.

Please find below our point-by-point responses to each of the comments.

Response to Reviewer 1 Comments

Comment 1-1.

Abstract

The abstract is lengthy but is lacking in key aspects. Primarily of concern, there is no actual data given for results. This is of absolute importance and the key statistical analyses and numerically reported outcomes must be presented.

Response 1-1:

Thank you for your insightful comment. We have revised the Abstract to include specific figures, such as P-values to enhance clarity and precision. We hope the revised abstract addresses the reviewer’s concern satisfactorily.

Comment 1-2.

Introduction

The second and third sentence begin with the word "It". Be more direct in your writing style.

Response 1-2:

Thank you for your valuable suggestion. In accordance with the suggestion, we have revised the sentence (lines 43–45 in the revised manuscript). In addition, the manuscript has been edited by the same professional English editing service as in the previous version. We hope the revised text now reads more naturally.

Comment 1-3.

Line 62: Should be no y after thirst.

Response 1-3:

Thank you for your suggestion. We have removed the letter ‘y’ in the revised manuscript (line 53 in the revised manuscript).

Comment 1-4.

You need to develop a proper problem statement.

Response 1-4:

Thank you for your valuable feedback. This comment made us realise that the original introduction lacked clarity and consideration. In response, we re-organised the latter half of the introduction and added a new Fig 1 to accurately convey the study concept and hypothesis. Additionally, we revised the entire Discussion section to ensure consistency with the updated introduction. We appreciate the opportunity to improve the manuscript.

Comment 1-5.

Line 66: Your hypothesis statement occurs before your purpose statement.

Response 1-5:

Thank you for your valuable comment. In accordance with your suggestion, we have completely re-organised the latter half of the Introduction section. We hope that the revised version meets the reviewer’s expectations.

Comment 1-6.

Methods

My main concern is the design of the study.

It is never explicitly stated how fluid intake was measured. For example, did patients drink out of metered containers and the change in mass or volume was assessed?

Response 1-6:

Thank you for your valuable comment. We apologize for not explicitly highlighting this aspect in the previous manuscript, despite its central importance to the study. In response to your feedback, we have revised the manuscript in lines 137 to 142 to address this point.

As a clinical recommendation within the facility, all older residents were advised to maintain a fluid intake of approximately 1,500 mL per day. Staff members, including nurses, therapists, and care workers provided fluids such as water or green tea in cups marked with a scale, both at the bedside and in the refectories during meals and upon request. The amount of fluid consumed was documented in the residents’ clinical records as a part of routine clinical practice. We have referenced these clinical records in our study.

Comment 1-7.

You have a within subjects design, but there seems to be no true control and intervention arm. First assessment was taken between 0-16 days after admission. Then, a second assessment occurred approximately 83 days later. The induction of blinded, reduced or increased fluid intake after a few weeks of hospitalization would have been a much stronger design. The ultrasonographic score changes with reduced or increased FI could have been confirmed, and then you could even have simply tested cognition during periods of lower and higher cererbral blood flow dynamics parameters.

Response 1-7:

Thank you for your insightful comment regarding our study design. We apologize for the insufficient description in the previous version. In accordance with your suggestion, we added a new Fig 1 that illustrates the study concept and protocol at a glance. We hope this addition adequately addresses your concerns.

One of the primary objectives of the current study is to examine the relationship between fluid intake and changes in cognitive state (Theme 1 in Fig 1). The analyses are based on correlation analyses and do not require a control group (lines 240–241 in the revised manuscript). Although a more robust design might involve randomized induction of fluid intake levels, such an approach would be ethically problematic, as increased fluid intake is recommended by the EFSA guidelines as part of standard clinical practice [1]. We aim to encourage participants to consume approximately 1,500 mL of fluid per day, aligning with clinical recommendations. Providing instructions to limit or alter fluid intake without clinical indication would be inappropriate and unethical. However, we remain vigilant about the potential adverse effects of excessive fluid intake, particularly concerning the risk of heart failure (details provided in the revised Introduction and the left panel of Fig 1B). This concern motivated the design of our study. Although we acknowledge that a randomized controlled approach might be preferable from a purely scientific perspective, we believe that the current study design is the most ‘practical’ and ethically acceptable within the clinical setting.

Comment 1-8.

No background or explanation of the cognitive status questionnaires is given.

Response 1-8:

Thank you for your valuable comment. In accordance with your suggestion, we have added the relevant description in lines 166–179 in the revised manuscript. The MMSE is the most widely used tool for dementia screening [2], primarily assessing learning and memory performance [3]. The FAB is another neuropsychological assessment designed to evaluate frontal lobe function concisely [4], an area in which the MMSE is less sensitive [5]. Both MMSE-J and FAB are scored on scales of 0–30 and 0–18, respectively; lower scores indicate more severe cognitive impairment. These two assessments were selected for the following reasons. First, the MMSE-J is routinely used in our facility as a part of standard clinical practice. Second, the clinical staff are familiar with the FAB, which is less time-consuming and more sensitive to certain cognitive domains that the MMSE may not effectively evaluate [6]. We hope that the revised manuscript satisfactorily addresses the reviewer's concerns.

Comment 1-9.

Number of participants: There are so many missing pieces of data it is difficult to follow the paper. All participants with missing data need to be removed. The abstract makes it appear that the n is almost double of the participants applicable in one of your most important figure (i.e. Figure 1) when the 3 outliers were removed. I am also not convinced the rationale for removing this data is valid.

Response 1-9:

Thank you for your helpful comment. In accordance with your suggestion, we have re-organised the entire manuscript—from the introduction to the discussion. We acknowledge that the original manuscript’s description was somewhat complex, which may have obscured the core objectives. Your feedback highlighted that the study's purpose was not clearly articulated and that essential and trivial information were intertwined. In the revised version, we summarised the main topic into two ‘Themes’, which are visually summarised in the new Fig 1. Simply put, the study was motivated by a clinical question: Should we encourage older individuals to increase fluid intake for better cognition without restrictions? (lines 70–73 in the revised manuscript) Clinicians often hesitate to promote increased fluid intake due to concerns about the risk of heart failure [lines 64–70 in the revised manuscript and the new Fig 1B]. Based on this, we hypothesized that excessive fluid intake might not enhance cognition, drawing an analogy from the relationship between blood volume and cardiac output. It is intuitively understandable that very high fluid intake does not improve MMSE-J scores, as demonstrated by the reversed U-shaped curve in the new Fig 2A (lines 410–413 in the revised manuscript).

In the revised manuscript, all analyses are categorized into three groups:

Theme 1: Comparison between fluid intake and changes in cognitive state

Theme 2A: Comparison between fluid intake and ultrasonographic parameters

Theme 2B: Comparison between cognitive state and ultrasonographic parameters

Theme 2B consists of three sub-comparisons (lines 255–256 in the revised manuscript).

Theme 2b-1: Comparisons at the first assessment

Theme 2B-2: Comparisons at the second assessment

Theme 2B-3: Comparisons of changes between the first and second assessments

(See in the revised Methods)

We eliminated analyses that were not central to our primary objectives.

These five analyses (Theme 1, 2A, 2B-1, 2B-2, and 2B-3) are essentially independent, and mismatches in the number of data points do not influence each other (lines XX–XX in the revised manuscript). Although it is possible to adjust all analyses to the dataset with the smallest number of data points, doing so would result in the loss of valuable information obtainable from the entire dataset. Thus, we chose to utilize the full available dataset, despite some missing data, except for the PI variable. We excluded PI from the revised manuscript due to significant missing data and because it does not provide additional important insights beyond those gained from the other analyses.

We hope we have correctly interpreted the underlying message behind comments 1–9 and have addressed it appropriately in the revised manuscript.

Comment 1-10.

No validation for the instrument used for determining BFP is provided. Did you see high variance in the patients? I may have overlooked where this outcome's data are located, but if no, why not just base relative fluid intake off body mass?

Response 1-10:

Thank you for clarifying regarding the LBM. In accordance with your suggestion, we have added the relevant description in lines 143–147. Distribution of bodily fluids varies across different body components: approximately three-fourths of the total body water is distributed within lean body mass, making it a more appropriate measure for evaluating body size [7–9]. The body fat percentage values were 27.9±8.2% (range: 10.9%–44.8%) at the first assessment and 31.0±7.6%(range: 9.8%–42.1%) at the second. We used a household-use body fat scale, categorised as a health meter. In Japan, the accuracy of such health matters is regulated by the Ministry of Economy, Trade and Industry. Additionally, all the body fat percentages were measured using the same device throughout the study to maintain consistency. Any potential measurement errors should be mitigated during the statistical analyses, even if the device has some inherent inaccuracies.

Comment 1-11.

There is just an overload of tables and figures. It is difficult to follow.

Response 1-11:

Thank you for pointing this out. We completely agree with the reviewer’s perspective and have re-organised the entire manuscript, from the Introduction section to the Discussion section.

We removed unnecessary analyses, figures, and tables, and focused on the data that support the main objectives of the study. The core concept of the study is now accurately illustrated in the revised Fig. 1 and is categorized into two themes. We hope that the revised manuscript effectively conveys our key messages.

Response to Reviewer 2 Comments

Comment 2-1.

Major comments.

The sample size is only 33 individuals, and they are all from a single geriatric health service facility. The sample has a high degree of homogeneity, which may lack broad representativeness. As a result, it is difficult to extrapolate the research findings to a more general elderly population. It is recommended that in future studies, the sample size be expanded and elderly people from different regions and with diverse life backgrounds be included.

Furthermore, It is impossible to assess the patients' fluid intake and dehydration status before admission, which may lead to biases in the research results. Although fluid intake was recorded during the study period, the situation before admission is unknown, which may interfere with the judgment of the relationship between fluid intake and cognition.

Response 2-1:

Thank you for your valuable comment. In accordance with your suggestion, we have revised the manuscript accordingly. With regard to the first point about the ‘homogeneity’ of study participants, we agree with the reviewer and have added this as a limitation in the manuscript (lines 504–507). Specifically, the critical amount of fluid intake is likely to vary among different groups of older individuals. If the present study is published, we plan to expand it by collaborating with other facilities within our hospital network and outside of our group, involving a larger number of participants to obtain more robust and generalizable results.

With regard to the amount of fluid intake prior to admission, we cannot entirely exclude its potential influence on the present results. However, we assume that the severity of dehydration prior to admission was randomly distributed across all participants, and our statistical analyses should have mitigated its effect. Although the uncontrolled levels of dehydration before admission may reduce the sensitivity of our analyses and increase the risk of type II error, we still observed significant correlations between mnFI and changes in cognition and other factors. Thus, believe this limitation is unlikely to substantially impact our main conclusions. We have added this explanation to the Limitations section (lines 507–515 in the revised manuscript).

Comment 2-2.

Abstract: The characteristics of the survey subjects is not well defined. It is unknown whether they are healthy, and if they have chronic diseases. During the research process, it is unclear how many subjects withdrew, and for what reasons. Also, the average age of the subjects and the duration of the study are not provided. The authors just showed the “several months”, three months? Six months? Please clarify.

Response 2-2:

Thank you for your helpful comment. Although we have already described some participant profiles, such as age, sex, and diagnosis, in Tables 1 and 2, we acknowledge that this information was not clearly organized and lacked a distinction between essential and non-essential information details. In the revised manuscript, we present a more concise and structured description of participant characteristics—such as age, sex, weight, lean body mass, and comorbidities—in the main manuscript (lines 96–101 and 276–283) as well as in Table 1. According to the study period (between the first and second assessments), it was 82.6 ± 14.9 days (from 34 to 119 days). Although this was mentioned in the previous version, it was not sufficiently prominent. In the revised manuscript, we have added a new Fig 1 illustrating the study concept and protocol, explicitly including the assessment interval (82.6 ± 14.9 days). Due to clinical limitations and constraints, the duration between assessments varied across participants. To maintain clarity, we have kept the expression ‘several months’ in several places throughout the manuscript.

A total of 33 participants were enrolled in the study, and none of them withdrew from the study. However, some data are missing due to clinical constraints—such as some participants not completing the second assessment by the data collection deadline (23 May 2023)

---

## [Editor Report · Decision Letter 1]

18 Sep 2025

Effect of fluid intake on cognitive function in older individuals: A prospective study

PONE-D-25-10914R1

Dear Dr. Shigihara,

We’re pleased to inform you that your manuscript has been judged scientifically suitable for publication and will be formally accepted for publication once it meets all outstanding technical requirements.

Kind regards,

Tanja Grubić Kezele, Ph.D., M.D.

Academic Editor

PLOS ONE
---

## [Editor Report · Acceptance letter]

PONE-D-25-10914R1

PLOS ONE

Dear Dr. Shigihara,

I'm pleased to inform you that your manuscript has been deemed suitable for publication in PLOS ONE. Congratulations! Your manuscript is now being handed over to our production team.

Kind regards,

on behalf of

Prof. dr. Tanja Grubić Kezele

Academic Editor

PLOS ONE